# THE QUEST FOR EFFICIENT REASONING: A DATA-CENTRIC BENCHMARK TO COT DISTILLATION

**Ruichen Zhang**[*1], **Rana Muhammad Shahroz Khan**[*1], **Zhen Tan**[2], **Dawei Li**[2], **Song Wang**[3],
**Tianlong Chen**[1]
[1]University of North Carolina at Chapel Hill,[2]Arizona State University,
[3]University of Virginia

## ABSTRACT

Data-centric distillation, including data augmentation, selection, and mixing, offers a promising path to creating smaller, more efficient student Large Language Models (LLMs) that retain strong reasoning abilities. However, there still lacks a comprehensive benchmark to systematically assess the effect of each distillation approach. This paper introduces **DC-CoT**, the first data-centric benchmark that investigates data manipulation in chain-of-thought (CoT) distillation from method, model and data perspectives. Utilizing various teacher models (e.g., `o4-mini`, `Gemini-Pro`, `Claude-3.5`) and student architectures (e.g., $3B$, $7B$ parameters), we rigorously evaluate the impact of these data manipulations on student model performance across multiple reasoning datasets, with a focus on in-distribution (IID) and out-of-distribution (OOD) generalization, and cross-domain transfer. Our findings aim to provide actionable insights and establish best practices for optimizing CoT distillation through data-centric techniques, ultimately facilitating the development of more capable reasoning models. The codebase can be accessed here.

## 1 INTRODUCTION

Large language models (LLMs) achieve strong reasoning performance when combined with *chain-of-thought* (CoT) prompting (Wei et al., 2022), but the best performance typically comes from expensive models with tens or hundreds of billions of parameters. To address it, *knowledge distillation* (KD) stands out to transfer reasoning skills to lighter students (e.g. 3–8 B) at low inference cost (Hinton et al., 2015; Ho et al., 2022; Mukherjee et al., 2023; Wang et al., 2022b). Among various KD strategies for CoT Xu et al. (2024); Tan et al. (2024), data-centric methods—such as augmentation, selection, and mixing—have gained popularity for being architecture-agnostic and cost-efficient Xu et al. (2023). However, a systematic assessment is still lacking to evaluate the effectiveness of these techniques.

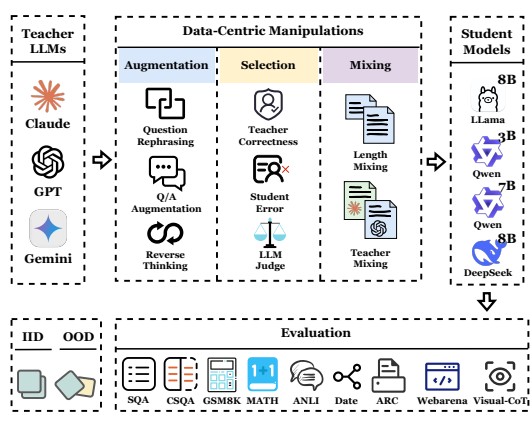

Figure 1: Overview of DC-CoT pipeline.

To address this, building a *data-centric* benchmark is essential. Such a benchmark will provide a clearer understanding of the performance of existing data-centric methods by systematically evaluating and answering fundamental questions, such as how to effectively synthesize, select, and mix various CoT samples to robustly boost the student models' performance. Furthermore, a data-centric benchmark will serve as a valuable and controlled evaluation resource for future research and the development of new techniques in this area. In this work, we introduce **DC-CoT**, the

---

[*] Equal contribution.

first benchmark designed to investigate data-centric CoT distillation systematically, answering the following research questions:

> ❶ **Method Perspective**: How can various data-centric CoT distillation methods for LLMs be categorized, and what is their comparative performance in enhancing student model reasoning?
>
> ❷ **Model Perspective**: How do the relative sizes and architectures of teacher and student models influence the effectiveness of data-centric CoT distillation?
>
> ❸ **Data Perspective**: How do different data characteristics and settings, such as in-distribution (IID) versus out-of-distribution (OOD) data, easy-to-hard generalization, and data availability, impact the outcomes of Chain-of-Thought distillation?

Regarding the Method Perspective, DC-CoT investigates various data manipulation strategies across three core axes: *(i) Augmentation*: Techniques like reverse reasoning and question/answer re-phrasing beyond vanilla CoT. *(ii) Selection*: Compare heuristics such as teacher-correct filtering, student-error prioritization, and LLM-based quality judges. *(iii) Mixing*: Explore blending CoT data based on length, domain, and teacher origin. To explore the Model Perspective, DC-CoT incorporates diverse teacher models (e.g., GPT-4o, Claude 3.5, Gemini-1.5-Pro) and various open-source student model families and sizes (e.g., LLaMA, Qwen, Gemma at 3-8B parameters). To address the Data Perspective, evaluations are conducted across reasoning datasets, specifically examining performance in in-distribution (IID) and out-of-distribution (OOD) settings.

Through extensive experiments, we present key findings and insights guided by research questions across multiple perspectives. From the Method Perspective, we find data augmentation to be generally the most effective approach and provide fine-grained analyses for each manipulation method across task types. From the Model Perspective, we confirm the roles of compatibility and learnability, highlighting their non-trivial impact on distillation and explaining why certain teacher–student pairs may fail. From the Data Perspective, we reveal distinct scaling behaviors across augmentation methods and quantify the generalization capabilities of student LLMs across datasets. All these insights will help guide future research toward more effective and efficient CoT distillation paradigms.

**In Summary**, our work makes the following contributions:

> ① We present DC-CoT, a unified, data-centric benchmark that explores data manipulation in distillation from method, model and data perspectives.
> ② We conduct extensive experiments across diverse teacher–student pairs, tasks, and datasets, offering the first large-scale empirical overview of CoT distillation.
> ③ We distill actionable guidelines—e.g., which augmentation boosts generalization, which filtering criterion balances quality and coverage, and when heterogeneous teacher mixtures help—thereby charting a path toward smaller yet more capable reasoning models.

## 2 RELATED WORKS

**Reasoning in LLMs.** Chain-of-Thought (CoT) elicits explicit intermediate reasoning steps, making LLM inference more transparent and markedly more accurate on multi-step tasks (Wei et al., 2022; Kojima et al., 2022). Based on this, newer *long-CoT* methods—e.g., Tree-of-Thought, iterative self-reflection, and self-correction—scale CoT by exploring multiple paths and refining answers through critique (Yao et al., 2023a; Madaan et al., 2023; Yu et al., 2025; Li et al., 2025c).

**Knowledge Distillation in LLMs.** Knowledge distillation transfers the behaviour of a large *teacher* LLM to a smaller, cheaper *student*. Beyond the original "soft-label" paradigm (Buciluǎ et al., 2006; Hinton et al., 2015), recent work treats LLM-generated instructions, responses, and rationales as synthetic supervision for supervised fine-tuning or alignment tuning (Kim et al., 2023; Tong et al., 2024; Ouyang et al., 2022; Zhang et al., 2024; Wang et al., 2024). A particularly effective variant is *reasoning* or chain-of-thought (CoT) distillation: instead of imitating only the final answer, the student is trained to follow the intermediate reasoning produced by the teacher, which has proved crucial when capacity or architectural gaps exist (Hsieh et al., 2023; Mukherjee et al., 2023; Lewkowycz et al., 2022; Yu et al., 2023). Despite promising gains, the field still lacks principled guidance on (i) which teachers, (ii) which rationales, and (iii) what selection or mixing strategies yield maximal

benefit for a given student, motivating a more data-centric exploration of CoT distillation. More detailed related work is given in Appendix D.

## 3 METHODOLOGY: A DATA-CENTRIC CoT DISTILLATION BENCHMARK

### 3.1 DATA-CENTRIC MANIPULATION

The central theme of our **DC-CoT** is the systematic evaluation of *data-centric manipulations* applied to CoT exemplars for knowledge distillation. These manipulations encompass various strategic operations to transfer the initial dataset $D^{source}$ to the target dataset $D^{target}$ for small student model training, potentially guided by a set of parameters or rules $\Theta$: $D^{target} = \mathcal{M}(D^{source}, \Theta)$. Here, $\mathcal{M}$ represents the abstract data transformation function encompassing augmentation, selection, and mixing. For augmentation strategies, we denote $L$ as the number of synthetic samples generated per source instance. **DC-CoT** is designed to deconstruct and analyze the impact of instantiating $\mathcal{M}$ through three primary types of data-centric operations: ❶ Data Augmentation (Section 3.1.1), ❷ Data Filtering (Section 3.1.2), and ❸ Data Mixing (Section 3.1.3).

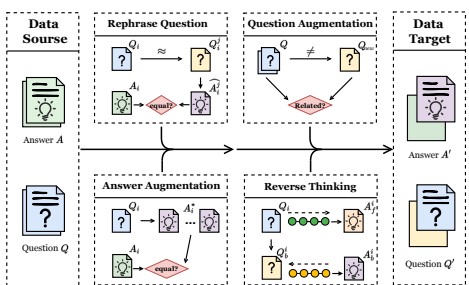

Figure 2: Data-centric augmentation flow. Teacher CoT traces are independently transformed by four operations: Rephrase Question, Question Augmentation, Answer Augmentation, and Reverse Thinking.

#### 3.1.1 DATA AUGMENTATION

Data Augmentation is crucial in CoT distillation by enriching and diversifying the training data ($D^{source}$) available for the student model, to expose the student to various reasoning patterns, question formulations, and explanatory styles for enhancing their reasoning capabilities and generalization. Within the DC-CoT benchmark, we investigate several data augmentation strategies as follows:

❶ **Question Rephrasing:** This method, introduced in MetaMath (Yu et al., 2023), aims to increase question diversity by having the teacher LLM $\mathcal{T}$ paraphrase an existing question $Q_i$ while preserving its underlying meaning and original answer $A_i^*$: $\{\hat{Q}_i^j = \mathcal{T}(Q_i, P_{reph})\}_{j=1}^L$. Here $Q_i$ and $P_{reph}$ are the original question and rephrasing prompt. For each rephrased question $\hat{Q}_i^j$, the teacher $\mathcal{T}$ generates a CoT rationale $\hat{R}_i^j$ and answer $\hat{A}_i^j$. one augmentation is retained if $\hat{A}_i^j$ matches the original answer.

❷ **Question Augmentation:** This strategy focuses on creating entirely new questions $Q_{new}$, to broaden the topical coverage or complexity of the training data, based on a set of seed questions $Q$ (Li et al., 2024a): $Q_{new} = \mathcal{T}(Q, P_{QA})$. $P_{QA}$ here is a prompt for generating novel questions. After that, the same generation-then-filter process will be adopted to produce new answers and CoTs for the augmented questions, as we introduced in the Question Rephrasing method. Unlike general instruction-tuning methods (e.g., Self-Instruct), this operation is strictly constrained to *Reasoning Transfer*. The prompt $P_{QA}$ forces the generation of parallel reasoning problems (e.g., altering numerical values in math or subjects in logic puzzles) to ensure the student learns the underlying reasoning pattern rather than memorizing specific answers.

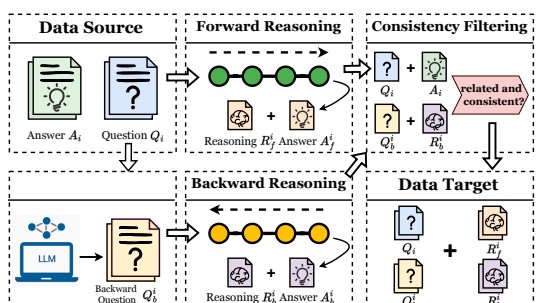

Figure 3: Reverse-Thinking augmentation pipeline: from each (question, answer) pair, generate forward reasoning, synthesize a backward question with its reasoning, then keep only examples whose forward-backward chains pass a consistency check.

❸ **Answer Augmentation:** It involves prompting the teacher LLM $\mathcal{T}$ to generate multiple diverse CoT rationales $R$ that all lead to the same correct ground-truth answer $A_i^*$ (Yu et al., 2023). Given $(Q_i, A_i^*) \in D^{source}$, and using a CoT generation prompt $P_{AA}$, the teacher model generates $L$

candidate rationales and answers as follows: $\{(R_i^k, A_i^k) = \mathcal{T}(Q_i, P_{AA}, \text{temp})\}_{k=1}^L$. To mitigate the risk of reasoning hallucinations, the prompt explicitly conditions the teacher on the ground-truth answer $A_i^*$. Our empirical results suggest that the benefit of exposing the student to diverse valid reasoning paths outweighs the noise of occasional imperfect traces, as the student learns the intersection of valid logic across the augmented set.

❹ **Reverse Thinking Augmentation** Reverse Thinking was introduced in the RevThink (Chen et al., 2024b). The goal is to enrich the data by generating forward CoT reasoning $R_f$, a corresponding backward question $Q^b$, and backward reasoning $R_b$. For each $(Q_i, A_i) \in D^{source}$ we do the following:

- *Generate Forward Reasoning:* $R_f^i = \mathcal{T}(Q_i, P_f)$ for some prompt $P_f$. This is filtered to ensure that the outcome of $R_f^i$ is the ground truth $A_i$.
- *Generate Backward Question:* Using a prompt $P_{bq}$, the teacher $\mathcal{T}$ generates a question that inverts the original problem: $Q_b^i = \mathcal{T}(Q_i, A_i, P_{bq})$.
- *Generate Backward Reasoning:* The teacher then generates the CoT for this backward question: $R_b^i = \mathcal{T}(Q_b^i, P_{br})$ for some prompt $P_{br}$.
- *Consistency Filtering:* A consistency check $c = \mathcal{T}(Q_i, A_i, Q_b^i, R_b^i, P_{con})$ is performed for making sure the backward and the forward questions are related and consistent with each other Yang et al. (2025). Only consistency quadruplets $(Q_i, R_f^i, Q_b^i, R_b^i)$ where $c = 1$ are retained.

### 3.1.2 DATA FILTERING

Data Filtering, or selection, is a critical step applied to either initial source data $D^{source}$ or augmented data to create a high-quality training set $D^{train}$ for the student model. Since not all CoT instances are equally beneficial, as some are noisy or incorrect, filtering aims to identify and retain the most valuable exemplars to optimize learning. Our DC-CoT investigates the following data selection strategies:

❶ **Filtering by Teacher Correctness:** This strategy used in (Ho et al., 2022), retains CoT instances where the teacher's final answer $A_i$ matches the ground-truth answer $A_i^*$: $D^{target} = \{(Q_i, R_i, A_i)|A_i = A_i^*\}$. This ensures the student learns from CoTs lead to correct outcomes.

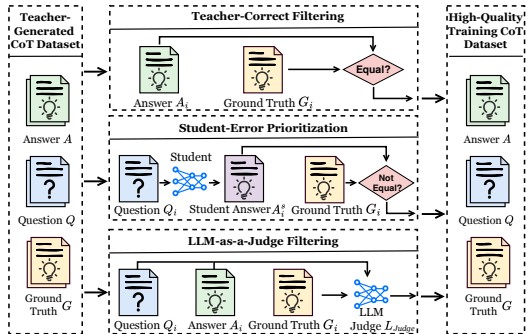

Figure 4: Data-filtering pipeline in DC-CoT. A teacher-generated CoT pool is refined through three selectors.

❷ **Filtering by Student Error:** This filtering strategy focuses student learning on its weaknesses by selecting instances where the student model yields an incorrect answer: $D^{target} = \{(Q_i, R_i, A_i)|\hat{A}_i \neq A_i^*\}$. This concentrated learning can focus on students' underperformed areas.

❸ **LLM-as-a-Judge Filtering:** Inspired by I-SHEEP (Liang et al., 2024), this method uses an external LLM $\mathcal{L}_{judge}$ to assess CoT instance quality based on criteria like coherence, correctness, and clarity, allowing for a nuanced quality assessment Li et al. (2024b; 2025a): $\text{Score}_i = \mathcal{L}_{judge}(A_i, R_i, Q_i, P_{eval})$. Instances are retained if their score meets a threshold $\tau$, making the final dataset become: $D^{source} = \{(Q_i, R_i, A_i)|\text{Score}_i \geq \tau\}$. To validate the reliability of this automated judge, we conducted a human evaluation on a random sample of 100 filtered instances from SQA and GSM8K. We observed a Cohen's Kappa ($\kappa$) of 0.84, indicating strong agreement between the LLM Judge (GPT-4o) and human experts, with the Judge exhibiting a slight preference for strictness—a desirable bias for high-quality distillation.

### 3.1.3 DATA MIXING

Beyond augmentation and selection, Data Mixing offers another avenue for data-centric manipulation in CoT distillation. This strategy involves strategically combining CoT instances from different distributions or with varying characteristics to create a more diverse training dataset $D^{target}$ for the student model. The core idea is that a blend of reasoning styles, complexities, or teacher provenances can lead to a student model with more robust and generalizable reasoning capabilities.

Table 1: Accuracy of augmentation, selection and mixing strategies on Llama-3.1-8B. Textual scores are the mean of three independent runs.

| | Agentic | Visual | | | Textual | | | | | | | AVG. | |
|---|---|---|---|---|---|---|---|---|---|---|---|---|---|
| | WebArena | Visual-CoT | OK-VQA | CLEVR | SQA | CSQA | ARC | MATH | GSM8K | ANLI | Date | Visual+Agentic | Textual |
| **Data Augmentation** | | | | | | | | | | | | | |
| Zero Shot | 5.66 | 42.10 | **65.60** | 56.88 | 57.64 | 43.08 | 48.46 | 9.32 | 19.64 | 33.83 | 49.70 | 23.88 | 37.38 |
| Zero Shot CoT | 8.25 | 44.52 | 61.84 | **58.36** | 65.55 | 53.56 | 67.41 | 11.76 | 21.00 | 39.92 | 62.13 | 26.39 | 45.90 |
| No CoT | 30.05 | **46.66** | 62.18 | 52.12 | 59.89 | 65.36 | 60.41 | 7.39 | 20.74 | 35.42 | 50.37 | **38.36** | 42.80 |
| Vanilla CoT | 22.78 | 45.44 | 59.94 | 54.04 | 58.08 | 69.37 | 55.63 | 4.38 | 24.30 | 23.92 | 57.02 | 34.11 | 41.81 |
| Rephrase Question | - | - | - | - | 59.73 | 62.95 | 67.01 | 16.52 | 38.86 | 42.47 | 59.41 | - | 49.56 |
| Question Aug | - | - | - | - | 60.40 | 61.47 | 70.37 | 20.31 | 44.03 | 41.26 | 61.07 | - | 51.27 |
| Answer Aug | - | - | - | - | 64.49 | 64.57 | 81.61 | 36.84 | 53.48 | 40.29 | 61.80 | - | 57.58 |
| Reverse Thinking | - | - | - - | - | 72.49 | 78.46 | 82.17 | 35.52 | 76.35 | 49.75 | 70.41 | - | **66.45** |
| **Data Selection** | | | | | | | | | | | | | |
| No Selection | 22.78 | 44.52 | 59.94 | 54.04 | 59.89 | 65.36 | 60.41 | **7.39** | 20.74 | **35.42** | 50.37 | 33.65 | 42.80 |
| Filtering with Teacher | 14.66 | 45.50 | 63.80 | **67.60** | **61.43** | 70.72 | **62.86** | 5.04 | **30.27** | 24.11 | 58.69 | 30.08 | **44.73** |
| Filtering with Student | 27.59 | 45.90 | **66.30** | 57.02 | 60.29 | **70.85** | 60.30 | 5.21 | 26.97 | 25.40 | 58.04 | **36.75** | 43.87 |
| Judge LLM | 15.64 | **46.54** | 59.42 | 54.12 | 54.83 | 62.49 | 57.46 | 3.43 | 22.72 | 26.51 | **59.85** | 31.09 | 41.04 |
| **Data Mixing** | | | | | | | | | | | | | |
| No Mixing | 22.78 | 44.52 | 59.94 | 54.04 | **59.89** | 65.36 | 60.41 | **7.39** | 20.74 | **35.42** | 50.37 | **33.65** | **42.80** |
| Length Mixing | - | - | - | - | 58.58 | **68.04** | 54.79 | 4.64 | **21.84** | 22.50 | **59.63** | - | 41.43 |
| Teacher Mixing | 21.18 | **45.48** | **61.7** | **55.6** | 56.75 | 66.94 | **62.82** | 5.96 | 19.57 | 29.46 | 52.30 | 33.33 | 41.97 |

❶ **Length-based CoT Mixing:** Length-based mixing, introduced in (Li et al., 2025b), combines CoT examples of varying reasoning length to help bridge this learnability gap for smaller models and offers complexity for larger models. This mix, controlled by a ratio $\alpha$, aims to provide a balanced curriculum, exposing students to detailed and concise reasoning.

❷ **Teacher-based CoT Mixing:** This method blends CoTs generated by different teachers (Li et al., 2025b). The mixed dataset is again guided by a ratio $\alpha$, providing a balanced set of reasoning examples and preventing smaller students from being overwhelmed while still offering sophisticated examples.

## 4    EXPERIMENT RESULT & ANALYSIS

### 4.1    BENCHMARK SETUP

**Teacher Models.** We use SoTA LLMs known for strong reasoning to generate CoT rationales: (1) Gemini-1.5-Pro (Team et al., 2024a), (2) GPT-4 (Achiam et al., 2023), (3) Claude-3.5 Sonnet (Anthropic, 2024), (4) GPT-4.1 mini (OpenAI, 2024a), (5) o4 mini (OpenAI, 2024b). Utilizing multiple teachers allows us to study the impact of teacher diversity. Data filtering is performed with task-specific Judge LLMs: LLama-2-70B for textual tasks, GPT-4o-mini (Achiam et al., 2023) for agentic tasks, and GPT-4/4.1-mini for visual tasks.

**Student Models.** We test these open-source models as students: (1) LLama-3.1-8B (Grattafiori et al., 2024), (2) LLama-3.1-8B-R1 Distilled (Guo et al., 2025), (3) Mistral-7B (Jiang et al., 2023), (4) Gemma-7B (Team et al., 2024b), and (5) Qwen-2.5-7B (Yang et al., 2024). **Baselines.** For Baseline comparison, we evaluate the models for (1) Zero Shot performance on the tasks, (2) Generate Zero-Shot CoT (Kojima et al., 2022), (3) Fine-tune the model on the dataset without any CoT, and (4) Vanilla CoT generated by the teacher model with no augmentation/filtering/mixing.

**Datasets.** Student performance is assessed on diverse reasoning datasets covering various skills and complexities. We evaluate textual reasoning tasks on: *Commonsense Reasoning Tasks:* StrategyQA (SQA; (Geva et al., 2021)), CommonsenseQA (CSQA; (Talmor et al., 2019)), ARC-challenge (ARC; (Clark et al., 2018)). *Math Reasoning:* GSM8K (GSM8K; (Cobbe et al., 2021)), MATH (MATH; (Hendrycks et al., 2021)). *Natural Language Inference:* ANLI (ANLI; (Nie et al., 2020)). *Logical Reasoning:* Date Understanding (Date; (Srivastava et al., 2022)). We evaluate agentic reasoning tasks on WEBARENA (Zhou et al., 2023), and evaluate visual reasoning on Visual-CoT (Shao et al., 2024), OK-VQA (Marino et al., 2019), and CLEVR (Johnson et al., 2017). We classify Shopping, Map, and Reddit as webarena-easy, and others as hard. For task descriptions, please refer to E.

### 4.2    METHOD-LEVEL RESULTS

This section delves into the performance of various data-centric manipulation strategies by posing key questions and deriving insights from our experimental findings. The analysis primarily references Table 1. It is important to note that the results discussed in Table 1 all pertain to the *Llama-3.1-8B* student model. Furthermore, the teacher model for visual tasks was *GPT-4-mini* (Achiam et al., 2023),

Table 3: Reverse-augmented distillation results for different teacher / student combinations on textual tasks; numbers are three-run averages.

| Student Model | Teacher Model | SQA | CSQA | ARC | MATH | GSM8K | ANLI | Date | AVG. |
|---|---|---|---|---|---|---|---|---|---|
| Llama-3.1-8B | Gemini-1.5-Pro | **72.49** | **78.46** | 82.17 | 35.52 | **76.35** | 49.75 | **70.41** | **66.45** |
| | GPT-4 | 70.74 | 71.93 | **83.64** | 34.60 | 70.72 | **51.37** | 68.51 | 64.50 |
| Llama-3.1-8B-R1 | Gemini-1.5-Pro | 69.43 | 71.74 | 74.23 | **36.82** | 69.45 | 47.08 | **70.41** | 62.74 |
| | GPT-4 | 70.95 | 68.40 | 76.84 | 36.27 | 70.94 | 50.58 | 67.80 | 63.11 |
| Mistral-7B | Gemini-1.5-Pro | 72.05 | 75.53 | 76.96 | 16.12 | 59.21 | 45.00 | 59.17 | 57.72 |
| | GPT-4 | 71.08 | 72.63 | 76.85 | 15.39 | 58.86 | 45.62 | 60.19 | 57.23 |
| Gemma-7B | Gemini-1.5-Pro | 68.12 | 74.86 | 73.46 | 16.54 | 53.45 | 40.92 | 31.36 | 51.24 |
| | GPT-4 | 69.08 | 73.81 | 75.60 | 16.18 | 54.49 | 41.65 | 30.57 | 51.63 |

for agentic tasks it was *Claude-3.5* (Anthropic, 2024), and for textual tasks, *Gemini-1.5-Pro-001* (Team et al., 2024a) was used. For the mixing, we use the models as described in Table 3 and 4.

**Q1: How do the broad categories of data-centric manipulation compare in terms of overall effectiveness?** Table 1 shows that Data Augmentation strategies yield the most substantial average performance uplift over the Vanilla CoT baseline. For instance, Reverse improves average accuracy on all eight tasks by $24.64\%\uparrow$. Filtering with Teacher Correctness (Textual Average: $44.7\%$) improves by $+1.93\uparrow$ over Vanilla CoT. The best mixing strategy, Teacher Mixing (Textual Average: $41.97\%$), shows a marginal decrease of $0.83\%\downarrow$ over Vanilla CoT. This confirms that for a moderately sized student (7-8B), creating diverse rationales is more impactful than selecting or reshuffling existing ones. Data selection is vital for quality control, and data mixing helps tailor its composition.

**Comparison with Logit-based Distillation.** While DC-CoT focuses on black-box distillation (where teacher logits are unavailable), we assessed its competitiveness against white-box methods using an open-weights teacher (Llama-3.1-70B) on the ARC-Challenge. As shown in Table 2, DC-CoT (Reverse Thinking) achieved $69.2\%$, significantly outperforming standard Logit-based KD $64.8\%$. This suggests that transferring explicit reasoning steps via data augmentation is more effective for reasoning tasks than minimizing divergence on the output distribution alone.

Table 2: Comparison of Data-Centric Distillation vs. Logit-based KD on ARC-Challenge (Teacher: Llama-3.1-70B).

| Method | Access Required | Accuracy (%) |
|---|---|---|
| Teacher Baseline | Weights/Logits | 92.4 |
| Standard KD (KL Div.) | Weights/Logits | 64.8 |
| Vanilla CoT (SFT) | Black-box (Text) | 60.4 |
| **DC-CoT (Reverse)** | **Black-box (Text)** | **69.2** |

**Q2: Which techniques are most effective for each data manipulation?** From Table 1, Reverse consistently excels, especially for structure logical deduction (MATH, GSM8K, Date). It likely fosters a deeper understanding by teaching bi-directional reasoning. Answer Augmentation also performs robustly, particularly for commonsense reasoning (SQA, CSQA), by exposing the student to varied solution paths, enhancing flexibility. While Question Augmentation and Rephrasing increase diversity, the more profound alterations from Reverse and Answer Augmentation generally yield larger gains. Among the selection techniques, LLM-as-a-Judge filtering is highly effective, often surpassing simpler heuristics due to its nuanced assessment of rationale quality (coherence, soundness) beyond mere answer correctness. However, filtering by Teacher Correctness is a strong baseline, ensuring students learn from factually accurate paths and consistently improve over no selection or other methods. When compared to the *No Mixing* baseline, data mixing strategies show varied effects. Length Mixing (Average: $41.43\%$) results in a slight decrease of $1.37\%\downarrow$ on average for textual tasks. However, while underperforming on others, it shows improvements on specific textual datasets like CSQA, GSM8K, and Date. Teacher Mixing also shows a slight decrease of $0.83\%\downarrow$ on average for textual tasks compared to *No Mixing*. These results suggest that the benefits of the tested mixing strategies are not universally additive over a strong *No Mixing* baseline for textual tasks on average, but they can offer advantages for specific datasets or modalities, likely by tailoring the data complexity or teacher style to particular student needs or task characteristics.

**Q3: Which data-centric methods show particular strengths for specific reasoning tasks?** Optimal strategies vary by task demands, and combining effective augmentation with suitable filtering or mixing can yield further improvements:

1. *Textual Reasoning (SQA, CSQA, ANLI):* Answer Augmentation and Question Rephrasing enhance linguistic diversity. These should be combined with LLM-as-a-Judge filtering to ensure the high

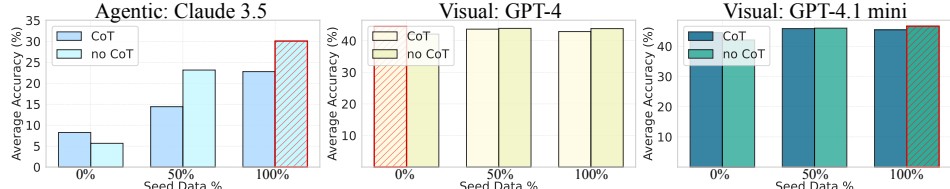

Figure 5: Accuracy of different seed-data sizes and teachers for WebArena and Visual-CoT. quality and coherence of the textual rationales. Teacher Mixing could also be beneficial after augmentation for tasks with varying teacher capabilities.

2. *Mathematical Reasoning (GSM8K, MATH, Date):* Reverse Thinking excels due to the need for backward deduction. Answer Augmentation is also valuable. These augmented datasets should then be rigorously filtered using Filtering by Teacher Correctness to eliminate any incorrect mathematical procedures. Subsequently, Length Mixing can be applied to balance the complexity of CoTs presented to the student.

3. *Agentic Reasoning (WebArena):* Given the complexity and potential for action chain errors, the augmented data should be curated using LLM-as-a-Judge filtering to enhance correctness.

4. *Visual Reasoning (Visual-Cot):* It is critical to use LLM-as-Judge filtering to ensure rationales are not only logically sound but also accurately reflect and reference the visual content.

### 4.3 MODEL-LEVEL RESULTS

We explored the effect of Teacher and Student types/sizes as well. For detailed results on Student Models, please refer to Appendix F.

#### 4.3.1 TEACHER MODEL ANALYSIS

We investigate the interplay between teacher and student models, summarized in Tables 3 and 4. For textual reasoning tasks, we utilize the best-performing augmentation approach, Reverse, and for visual as well as agentic tasks, we report the performance on vanilla CoT.

**Q4. How does the choice of a teacher model impact the performance of different student models on textual reasoning tasks? Is there a universally "best" teacher for all students?** Table 3 reveals that for textual reasoning, stronger models like Gemini-1.5-Pro and GPT-4 generally yield better results when distilling to capable student models such as Llama-3-8.1 B. For instance, Llama-3.1-8B achieves a high average textual score for both teachers, suggesting that as long as the teacher is powerful enough and the student has adequate capacity, transferring complex reasoning using Knowledge Distillation is quite effective. However, a universally "best"

Table 4: Impact of teacher model on agentic (WebArena) and visual (Visual-CoT) performance.

| Student Model | Teacher Model | WebArena | Visual-CoT |
|---|---|---|---|
| Llama-3.1-8B | Claude-3.5 | 22.78 | - |
| | GPT-4o | **24.51** | - |
| Llama-3.1-8B-R1 | Claude-3.5 | 11.33 | - |
| | GPT-4o | 13.79 | - |
| Qwen-2.5-VL-3B | GPT-4 | - | 42.92 |
| | GPT-4-mini | - | **45.44** |
| | o4-mini | - | 45.20 |

Table 5: Performance of Llama-3.1-8B and Mistral-7B when varying the percentage of seed data.

| Student Model | Seed Data % | Augmentation Type | SQA | ARC | GSM8K | Date | AVG. |
|---|---|---|---|---|---|---|---|
| Llama-3.1-8B | Zero-Shot | None | 57.64 | 48.46 | 19.64 | 49.70 | 43.86 |
| | 25% | Vanilla CoT | 68.12 | 79.95 | 42.99 | 66.86 | 64.48 |
| | | Reverse | 60.70 | 77.82 | 30.02 | 74.56 | 60.78 |
| | 50% | Vanilla CoT | **73.80** | 80.12 | 36.39 | 71.01 | 65.33 |
| | | Reverse | 62.88 | 79.95 | 47.01 | 68.64 | 64.62 |
| | 75% | Vanilla CoT | 67.69 | 71.78 | 26.61 | 65.89 | 57.99 |
| | | Reverse | 68.12 | 80.79 | 59.67 | **73.96** | 70.64 |
| | 100% | Vanilla CoT | 58.08 | 55.34 | 24.30 | 59.41 | 49.28 |
| | | Reverse | 72.49 | **82.17** | **76.35** | 70.41 | **75.36** |
| Mistral-7B | Zero-Shot | None | 55.02 | 50.94 | 20.24 | 46.75 | 43.24 |
| | 25% | Vanilla CoT | 70.46 | 69.52 | 44.09 | 63.58 | 61.91 |
| | | Reverse | 71.18 | 73.98 | 54.13 | 62.72 | 65.50 |
| | 50% | Vanilla CoT | 64.91 | 70.04 | 39.25 | 58.53 | 58.18 |
| | | Reverse | 68.56 | 76.11 | 53.90 | **64.59** | 65.79 |
| | 75% | Vanilla CoT | 62.14 | 64.02 | 26.94 | 50.69 | 50.95 |
| | | Reverse | 71.98 | **77.30** | 54.44 | 61.41 | 66.28 |
| | 100% | Vanilla CoT | 60.84 | 51.40 | 19.55 | 46.41 | 44.55 |
| | | Reverse | **72.05** | 76.96 | **59.21** | 59.17 | **66.85** |

teacher is not apparent. While Gemini-1.5 shows a slight edge for LLama-3.1-8B on average, GPT-4 can be comparable or better on specific datasets (e.g., ARC for Llama-3.1-8B). For Mistral Gemini-1.5, it slightly outperforms GPT-4, whereas for Gemma-7B, GPT-4 is marginally better than the other. This variability indicates that optimal teacher-student pairings are nuanced, likely influenced by factors like architectural alignment or specific knowledge domains.

**Q5. What does performance on agentic and visual tasks indicate about teacher model suitability?** Table 4, which examines agentic and visual tasks, provides strong support for the small

model learnability gap. This concept suggests that smaller student models (e.g., $\leq$ 3B parameters) may not always benefit most from the largest available teachers, as they might learn more effectively from slightly smaller teachers whose reasoning complexity better matches their own capacity. Our results for the Qwen-2.5-VL-3B student on Visual-CoT clearly demonstrate this: distillation from smaller, capable teachers like GPT-4-mini (45.44% acc.) and o4-mini (45.20% acc.) leads to superior performance when compared to the largest GPT-4 (42.92% acc.). This implies that the CoTs from very large models like GPT-4 might be overly complex for a smaller, specialized model like Qwen-2.5-VL-3B to internalize effectively. The more digestible reasoning patterns of GPT-4-mini and o1-mini likely facilitate better knowledge transfer, highlighting that sheer teacher strength does not guarantee optimal distillation if the student struggles with the complexity. **Q6. Considering textual, agentic, and visual tasks, what general principles can be inferred for selecting an optimal teacher?** Several interesting observations lead to emerging principles: *(1) The "Learnability Gap" Affects Smaller/Specialized Students.* For smaller or specialized students, the strongest teacher is not always the best. A teacher with more aligned reasoning complexity, even if smaller, can yield better results. *(2) Student's Prior Distillation History Impacts Receptiveness.* The Llama-3.1-8B-R1 model, previously distilled from DeepSeek-R1, shows slightly lower average performance on textual tasks compared to base Llama-3.1-8B when further distilled by either Gemini-1.5-Pro or GPT-4. This suggests that a student's prior specializations or distillation experiences can hinder learning from new teachers if their strengths don't align, leading to less effective knowledge transfer.

### 4.3.2 STUDENT ARCHITECTURE AND ADVANCED SELECTION

While we discuss the scaling laws of standard dense student models in Appendix F, it is crucial to validate the universality of DC-CoT across diverse architectures and assess more complex data selection heuristics. To this end, we extended our evaluation to **DeepSeek-VL2** (a Mixture-of-Experts model) and **Qwen-2.5-VL-8B** on visual reasoning tasks (OK-VQA, CLEVR). Furthermore, we introduced an **Uncertainty-based Selection** strategy, which prioritizes training instances where the student model exhibits high entropy ($> 0.5$) in zero-shot inference.

Table 6: Performance on Visual Reasoning tasks across Dense (Qwen) and MoE (DeepSeek) architectures.

| Model | Qwen-2.5 VL 8B (Dense) | | DeepSeek-VL2 (MoE) |
|---|---|---|---|
| Dataset | OK-VQA | CLEVR | OK-VQA |
| *Data Augmentation* | | | |
| Zero Shot | 65.60 | 56.88 | 11.60 |
| Zero Shot CoT | 61.84 | 58.36 | 12.92 |
| Vanilla CoT | 59.94 | 54.04 | 45.46 |
| *Data Selection* | | | |
| No Selection | 59.94 | 54.04 | 45.46 |
| Teacher Filter | 63.80 | **67.60** | **51.82** |
| Student Filter | **66.30** | 57.02 | 43.46 |
| LLM Judge | 59.42 | 54.12 | 43.88 |
| Model Uncertainty | 59.54 | 50.26 | 43.78 |
| *Data Mixing* | | | |
| No Mixing | 59.94 | 54.04 | 45.46 |
| Teacher Mixing | 61.70 | 55.60 | 48.04 |

As presented in Table 6, DC-CoT strategies remain effective for the MoE architecture. For instance, *Teacher Filtering* improves DeepSeek-VL2's performance on OK-VQA from 45.46% (Vanilla) to 51.82%. Regarding data selection, while Uncertainty-based selection yields competitive results (e.g., 59.54% on OK-VQA with Qwen), it does not consistently outperform our proposed heuristic methods (Student/Teacher Filtering). This suggests that the foundational primitives defined in DC-CoT are both robust and efficient for diverse student architectures including MoEs.

## 4.4 DATA-LEVEL RESULTS

### 4.4.1 EFFECT OF DATA VOLUME

We investigate the relationship between the volume of seed data used for distillation and the resulting student model performance, referencing Table 5 for the textual reasoning task with Gemini-1.5-Pro as the teacher and Reverse augmentation, and Figure 5 for agentic and visual tasks with Claude 3.5 as the teacher and CoT.

**Q7. How does increasing the percentage of seed data generally impact student model performance for Vanilla CoT and Reverse on textual tasks? How do these two methods compare?** On textual tasks, increasing seed data for Vanilla CoT does not consistently yield linear performance improvements. For Llama-3.1-8B, Vanilla CoT performance peaks at 50% seed data, then declines. Mistral with Vanilla Cot shows a similar non-linear trend, peaking earlier at 25%. This suggests that additional raw teacher traces might introduce noise or less informative examples beyond an optimal point, potentially hindering learning. In contrast, Reverse augmentation generally shows more consistent benefits with increased data. For both models, Reverse results in better performance

at higher data volumes. This indicates that the richer signal from Reverse is more effectively leveraged as data volume increases. **Q8. Does the "more data always leads to better results" scaling law hold true across these experiments?** The traditional scaling law does not universally hold in our experiments. This is particularly evident for Vanilla CoT on textual tasks, where performance can degrade with excessive data. However, more data tends to be beneficial up to the tested volumes for more sophisticated augmentations like Reverse on textual data, and generally for agentic tasks.

### 4.4.2 GENERALIZATION CAPABILITY ANALYSIS

We investigate how well reasoning skills learned through CoT distillation on a source dataset transfer to related but distinct target datasets. The analysis primarily references Table 4, while all experimental settings are explained in Appendix C.3.

Table 7: Zero-shot (ZS) versus OOD fine-tuning accuracy with Llama-3.1-8B.

| Training Data | Testing Data | Setting | ACC. |
|---|---|---|---|
| SQA | BoolQ | ZS | 54.75 |
| | | OOD | **64.16** |
| ARC | OBQA | ZS | 74.58 |
| | | OOD | **81.60** |
| ANLI | ESNLI | ZS | 49.74 |
| | | OOD | **59.75** |
| GSM8K | GSM8K-Rev | ZS | 16.74 |
| | | OOD | **38.89** |
| | MATH | ZS | **9.32** |
| | | OOD | 8.75 |
| MATH | GSM8K | ZS | 19.64 |
| | | OOD | **80.74** |
| Webarena-hard | Webarena-easy | ZS | 14.18 |
| | | OOD | **19.90** |
| Webarena-easy | Webarena-hard | ZS | 2.44 |
| | | OOD | **11.95** |
| Visual-CoT | OK-VQA | ZS | **42.10** |
| | | OOD | 38.90 |
| OK-VQA | Visual-CoT | ZS | 44.52 |
| | | OOD | **44.62** |

**9. How does fine-tuning on a source dataset generally impact Out-of-Distribution (OOD) performance compared to Zero-Shot performance on the target dataset?** Table 7 consistently shows that fine-tuning on a source dataset, even if different from the target, generally leads to substantial improvements in OOD performance on the target dataset compared to its Zero-Shot accuracy. For instance, after training on SQA, OOD performance on BoolQ improves. Similarly, training on ARC boosts OBQA performance. This trend holds across textual, mathematical, agentic, and even some visual task pairings, indicating that the reasoning skills learned via CoT distillation possess a notable degree of transferability. **Q10. Are there specific task categories or pairings where OOD generalization is particularly strong or weak? Does fine-tuning on a source task always guarantee better OOD performance than its Zero-Shot counterpart on the target task?** The degree of generalization varies across task categories and specific pairings as observed in Table 7. Strong generalization is evident when transferring between similar textual reasoning tasks. For example, training on SQA significantly boosts BoolQ, and ARC training enhances OBQA performance. Mathematical reasoning also shows strong positive transfer, particularly when training on the more complex MATH dataset and testing on GSM8K, and also from GSM8K to its reversed version, GSM8K-Rev. Agentic tasks within WebArena also demonstrate good generalization across difficulty levels. However, generalization can be mixed or weak in other scenarios. For instance, while MATH to GSM8K is strong, the reverse (GSM8K to MATH) shows a decrease. Visual tasks also present varied results; training on OK-VQA improves Visual-Cot, but training on Visual-Cot leads to a drop on OK-VQA.

For a detailed analysis of the computational efficiency and token-level costs of our data-centric pipeline, please refer to Appendix G.

## 5 CONCLUSION

This paper addresses the challenge of transferring reasoning from large to small models via CoT distillation, a domain where data-centric strategies have been underexplored. We introduce **DC-CoT**, a comprehensive benchmark designed to systematically investigate how data augmentation, selection, and mixing influence CoT distillation efficacy. Our findings reveal that data-centric manipulations significantly enhance distillation. Data augmentation, in particular, offers the most substantial performance gains by enriching the diversity of reasoning traces. Furthermore, we distill our findings into a heuristic framework for practitioners: (1) **Structured Logic tasks** (Math, Code) benefit most from **Reverse Thinking** combined with **Teacher Correctness** filtering to enforce logical consistency. (2) **Open-Ended Linguistic tasks** (Commonsense, NLI) require **Answer Augmentation** paired with **LLM-as-a-Judge** to capture diverse reasoning paths without semantic drift. (3) **Agentic and Visual tasks** necessitate **LLM-as-a-Judge** filtering, as simple heuristics fail to verify the grounding of rationales in observation contexts. Future work will expand this benchmark to include non-Transformer architectures and investigate more complex selection metrics, paving the way to democratize advanced reasoning.

ACKNOWLEDGEMENT

This work is generously supported by Amazon Research Award, Cisco Faculty Award, UNC Accelerating AI Awards, NAIRR Pilot Award, OpenAI Researcher Access Award, and Gemma Academic Program GCP Credit Award.

ETHICS STATEMENT

We adhere to the ICLR Code of Ethics. No private, sensitive, or personally identifiable data are involved. Our work does not raise foreseeable ethical concerns or produce harmful societal outcomes.

REPRODUCIBILITY STATEMENT

Reproducibility is central to our work. All datasets used in our experiments are standard benchmarks that are publicly available. We provide full details of the training setup, model architectures, and evaluation metrics in the main paper and appendix. We have also released our codebase as anonymous repository, including scripts for preprocessing, training, and evaluation, along with configuration files and documentation to facilitate exact reproduction of our results. Random seeds and hyperparameters will also be included to further ensure reproducibility.

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

## A   LIMITATIONS

- **Budget Constraints:** Due to budget constraints, models like GPT-o4 were not included in our experiments. Moreover, migrating to other benchmarks also incurs substantial API costs. Therefore, for agentic task similar to many related papers Zhou et al. (2023), we focus solely on the WEBARENA Zhou et al. (2023) framework. However, our method is simple and efficient, without any benchmark-specific optimizations, making it easily transferable to other models.

- **Hardware and Time Constraints:** Extending distillation to more and larger models is highly challenging due to hardware and time limitations. Therefore, we selected some student models for our distillation experiments.

## B   BROADER IMPACT

The DC-CoT benchmark is poised to significantly impact AI by fostering the development of smaller, more accessible, and powerful reasoning models. By systematically evaluating data-centric CoT distillation strategies, DC-CoT offers crucial insights and a standardized testbed, steering research towards resource-efficient AI and enabling advanced reasoning in computationally constrained environments. This research can yield broad societal and technological benefits:

1. *Democratization of AI:* Lowering computational barriers allows wider access to innovate with state-of-the-art AI.

2. *Educational Advancements:* Accessible reasoning models can be integrated into educational tools, supporting personalized learning.

3. *Application of AI:* Broader deployment of reasoning AI can aid complex problem-solving in research, healthcare, finance, and other industries.

The insights from DC-CoT will also guide practitioners in optimizing distillation pipelines, promoting data-aware and sustainable AI by reducing the computational footprint of large models. By facilitating more efficient reasoning systems, DC-CoT contributes to a future of more equitably accessible and sustainably developed advanced AI.

## C   EXPERIMENT SETTING

### C.1   DISTILLATION TRAINING

We conduct the distillation training on **8 A100 GPUs** and 16 **A6000 GPUs**, using LoRA fine-tuning for the student models. The LoRA rank we set is 32, and the lora alpha we set is 64. For an agentic task, the training process spans **5 epochs**, with a learning rate of $5 * 10^{-5}$ and a context length of $10,000$. For the visual task, the training process spans **1 epoch**, with a learning rate of $5 * 10^{-5}$. The distillation methodology follows the guidelines provided in `Llama Factory`(Zheng et al., 2024). For Textual tasks, we train for 10 epochs for each dataset.

### C.2   INFERENCE PIPELINE

For inference, we employ the `vLLM` framework, running on **8 A100 GPUs**. The WEBARENA framework is deployed on **4 CPU machines**. To enhance efficiency, we leverage the official task-parallel Bash script for parallel execution, rather than processing tasks sequentially by task ID.

### C.3   EXPERIMENTAL SETTINGS FOR GENERALIZATION EXPERIMENT

For all experiments, we use Llama-3.1-8B as our student model. OOD Datasets were chosen as follows: BoolQ (Clark et al., 2019) was used for SQA, OBQA (Mihaylov et al., 2018) for ARC, ESNLI (Camburu et al., 2018) for ANLI, GSM8K-Rev (Guo et al., 2024) and MATH for GSM8K, GSM8K for MATH, Webarena-easy for Webarena-hard and vice versa, Ok-VQA for Visual-CoT and vice versa.

# D RELATED WORK

## D.1 REASONING IN LLMs

The ability of LLMs to perform complex reasoning has been significantly enhanced by techniques that encourage explicit, step-by-step thinking. Foremost among these is Chain-of-Thought (CoT) prompting (Wei et al., 2022; Kojima et al., 2022; Nye et al., 2021), which elicits intermediate reasoning steps from LLMs before arriving at a final answer. This approach makes the model's inference process more transparent by providing human-readable explanations (Joshi et al., 2023; Lanham et al., 2023) and substantially improves performance on tasks requiring multi-step deduction, such as arithmetic, commonsense, and symbolic reasoning (Wei et al., 2022). By breaking down complex problems into manageable intermediate computations, CoT helps LLMs navigate intricate logical pathways and arrive at more accurate conclusions (Madaan & Yazdanbakhsh, 2022; Wang et al., 2023a; Dziri et al., 2023). Integrating self-generated rationales through CoT effectively boosts the reasoning capabilities inherent in these models (Kojima et al., 2022).

Building upon the foundational CoT paradigm, recent research has explored more sophisticated "deep-thinking" or "long-CoT" approaches to push the boundaries of LLM reasoning further. These methods often involve generating more elaborate or structured reasoning pathways. For example, Tree-of-Thought (Yao et al., 2023a) prompting allows models to explore multiple reasoning paths in parallel, evaluating intermediate thoughts to decide the most promising direction. Other techniques focus on iterative refinement (Wang et al., 2022b) and self-correction, such as Self-Reflection (Madaan et al., 2023; Yao et al., 2023b; Shinn et al., 2023), where models critique and improve their own generated thoughts.

## D.2 KNOWLEDGE DISTILLATION IN LLMs

Knowledge distillation is a potent technique for transferring knowledge from a large, often cumbersome, "teacher" model to a smaller, more efficient "student" model. This process is increasingly relevant in the context of LLMs due to their substantial size and computational demands. The fundamental concept, as introduced in early works (Buciluǎ et al., 2006; Hinton et al., 2015), involves training the student model to mimic the teacher model's output distribution (soft labels), thereby minimizing the divergence between their respective distributions. This approach has found applications across various tuning techniques for LLMs. For instance, LLM-generated annotations, including instructions, responses, and rationales, are leveraged in supervised fine-tuning, *i.e.*, where a smaller model learns from the synthetic data produced by a larger teacher LLM (Kim et al., 2023; Tong et al., 2024; Huang et al., 2023; Wang et al., 2023b; 2025; Lu et al., 2023). This is particularly useful for enhancing specific capabilities (Josifoski et al., 2023; Zhang et al., 2023; Zhao et al., 2023) or imparting domain-specific knowledge efficiently (Taori et al., 2023; Xu et al., 2023; Zheng et al., 2023; Wang et al., 2022a). Furthermore, distillation techniques are employed in alignment tuning. One example includes Reinforcement Learning from Human Feedback (RLHF) (Ouyang et al., 2022), where synthetic data from LLMs can aid in reward modeling and policy training to align model outputs with human preferences and intentions.

While classical knowledge distillation learns from the teacher model's distributions, and the objective is to minimize the difference between the student's distribution and the teacher's (Chen et al., 2020), recent advancements in LLMs have brought a particular focus to distilling their complex reasoning capabilities, especially CoT processes, into smaller student model's (Kojima et al., 2022). CoT is also crucial when addressing architectural differences or significant capacity gaps between teacher and student LLMs, as merely mimicking the final output might be insufficient for the student to learn effectively. Teacher models provide CoT rationales in various ways: (1) Sampled directly from the teacher (Hsieh et al., 2023; Fu et al., 2023; Li et al., 2023a; West et al., 2022; Magister et al., 2023; Mukherjee et al., 2023; Mitra et al., 2023), (2) Generated via bootstrapping (Li et al., 2023b; Ding et al., 2024; Zelikman et al., 2022; Lewkowycz et al., 2022; Yu et al., 2023; Li et al., 2024a; Yuan et al., 2023; Guo et al., 2024; Chen et al., 2024b), or (3) Obtained via multiple teacher models (You et al., 2017; Chen et al., 2024a). The rationale, reflecting the detailed thought process and reasoning pathway, serves as valuable auxiliary information for the student model to predict the final answer more accurately and robustly. While CoT distillation shows promise (Mukherjee et al., 2023; Ho et al., 2022), it remains unclear which methods, teacher models are most effective for a specific student model and how they perform in various settings. This calls for a data-centric study

of how the generation, selection, and combination of distillation data impact student reasoning and generalization.

## E   TASK DESCRIPTIONS

**Textual Reasoning:** It assesses a model's ability to make logical inferences from text, often through multi-step reasoning. Each instance includes a question $Q$, rationale $R$, and answer $A$. The student model $\mathcal{S}_\theta$ learns to predict $A$ using $Q$ and $R$. Tasks span commonsense, science, math, and table reasoning, with performance measured by answer accuracy.

**Agentic Reasoning:** This task tests an LLM agent $\pi_\theta$ in the WEBARENA browser sandbox, where it must follow an instruction $I$ by navigating real websites. At each step, the agent observes $o$, takes an action $a$, and explains its reasoning $r$. A large LLM ($M_L$) selects actions based on the interaction history. Performance is measured by Success Rate (SR)—the fraction of tasks where the agent reaches the correct goal state.

**Visual Reasoning:** Extends chain-of-thought to multi-modal inputs, requiring models to interpret visual content and answer related questions. Each instance is a tuple $(v, q, a, r)$: an image $v$, a question $q$, an answer $a$, and a rationale $r$ outlining reasoning steps linking $v$ to $a$. Unlike text-only reasoning, visual reasoning demands interpretable grounding—$r$ often points to specific image regions that justify the answer. This keeps the reasoning process transparent, testing the model's ability to connect visual cues with logical steps across multiple reasoning hops.

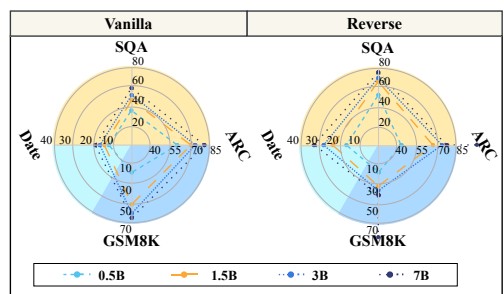

Figure 6: Qwen-2.5 (0.5B-7B) distilled with Vanilla-CoT vs Reverse

## F   IMPACT OF STUDENT MODEL SIZE

This section examines how the scale of the student model influences the efficacy of CoT distillation, with a particular focus on learnability from different augmentation strategies. The experiments, summarized in Figure 6, are performed on Qwen-2.5 models of varying sizes (0.5B, 1.5B, 3B, 7B) when distilled with Vanilla CoT and Reverse augmentation, using Gemini-1.5-Pro as the teacher.

**Q11. How does the student model size generally affect reasoning performance with standard Vanilla CoT, and how does it interact with more complex augmentations like Reverse?** As shown in Figure 6, performance with Vanilla CoT clearly scales with student model size: Qwen-2.5-0.5B achieves an average of $32.86\%$, which improves to $45.72\%$ for 1.5B, $50.89\%$ for 3B, and $55.58\%$ for the 7B model. This confirms that larger models better leverage standard teacher CoTs. The introduction of Reverse presents a more nuanced picture. On average across all four textual tasks, the impact is mixed; the 1.5B model shows a modest gain, while others see slight average decreases. However, these averages mask strong task-specific effects. Reverse significantly boosts performance on SQA and Date for all student sizes. Conversely, it markedly degraded performance on ARC and GSM8K compared to Vanilla CoT. This demonstrates that the utility of this complex augmentation is highly task-dependent in our specific student-teacher setup, instead of being a universal benefit.

**Q12. Do smaller student models (0.5B, 1.5B) exhibit the small model learnability gap when faced with complex augmentations like Reverse?** The small model learnability gap suggests smaller models struggle with overly complex reasoning. Analyzing our results: On tasks where Reverse is beneficial, smaller models (0.5B, 1.5B) achieve substantial gains. However, their absolute scores remain below those of larger students, indicating a capacity limitation in reaching peak performance.

# G  EFFICIENCY AND CODE ANALYSIS

Efficiency is an important consideration for a data-centric pipeline designed for broad adoption. While wall-clock time can be a useful metric, it often varies significantly depending on the hardware, batch sizes, and API latencies. To offer a more hardware-independent and reproducible measure of computational cost, we instead report token-level costs for data generation. The token usage for several key prompting techniques is summarized in the Table 8.

Moreover, these costs are incurred once during data generation, with no inference-time overhead or change to the student model architecture. This design choice was deliberate: we aimed to make DC-CoT practical for both academic and applied ML use cases.

Table 8: Token-level cost comparison for data generation methods.

| Method | Prompt Type | Avg. Prompt Tokens | Avg. Output Tokens | Total Tokens per Sample |
|---|---|---|---|---|
| Standard CoT | Forward CoT Prompt | 60 | 180 | 240 |
| Rephrased CoT | Question Rewriting | 75 | 180 | 255 |
| Reverse Thinking | Answer-First Reverse CoT | 110 | 200 | 310 |

To quantify the computational benefits of DC-CoT, we measured the throughput of our distilled students on a single NVIDIA A100-80GB GPU using vLLM. As shown in Table 9, the distilled Qwen-2.5-3B model—which achieves performance competitive with larger vanilla baselines—offers a $\sim$6–9$\times$ speedup relative to the 8B baseline, validating the "Efficient Reasoning" claim of our benchmark.

Table 9: Efficiency profile of distilled student models measured on A100-80GB.

| Model | VRAM (GB) | Throughput (tok/s) | Relative Speedup |
|---|---|---|---|
| Llama-3.1-8B (Student) | 16.2 | 115.4 | 1.0$\times$ |
| Qwen-2.5-3B (Student) | 7.8 | 184.2 | $\sim$1.6$\times$ |
| Qwen-2.5-1.5B (Student) | 4.2 | 245.1 | $\sim$2.1$\times$ |

# H  CONFIDENCE INTERVALS AND SIGNIFICANCE TESTING

While we reported average accuracy over 3 seeds for all experiments, we acknowledge that confidence intervals help contextualize gains that appear small. Some of these results are summarized in the Table 10. We report 95% confidence intervals for core reasoning tasks (ARC, MATH, GSM8K) across Mistral-7B, LLaMA-3.1-8B, and Gemma-7B models. As shown below, Reverse CoT consistently and significantly outperforms No CoT, with non-overlapping intervals in nearly all cases, confirming the robustness of the gains. These results suggest that improvements are statistically significant, not noise.

Table 10: Accuracy with 95% confidence intervals on core reasoning tasks.

| Model | Task | Method | Accuracy ± CI |
|---|---|---|---|
| Mistral-7B | ARC | No CoT | 68.26 ± 0.75 |
| | | Reverse CoT | 76.96 ± 1.45 |
| | MATH | No CoT | 7.98 ± 0.39 |
| | | Reverse CoT | 16.12 ± 0.38 |
| | GSM8K | No CoT | 31.11 ± 1.80 |
| | | Reverse CoT | 59.21 ± 0.85 |
| LLaMA-3.1-8B | ARC | No CoT | 60.41 ± 1.37 |
| | | Reverse CoT | 82.17 ± 1.20 |
| | MATH | No CoT | 7.39 ± 0.13 |
| | | Reverse CoT | 35.52 ± 0.26 |
| | GSM8K | No CoT | 20.74 ± 1.04 |
| | | Reverse CoT | 76.35 ± 1.98 |
| Gemma-7B | ARC | No CoT | 68.09 ± 1.17 |
| | | Reverse CoT | 73.46 ± 1.06 |
| | MATH | No CoT | 7.24 ± 0.34 |
| | | Reverse CoT | 16.54 ± 0.83 |
| | GSM8K | No CoT | 26.22 ± 0.83 |
| | | Reverse CoT | 53.45 ± 1.75 |

## I  THINKING EXAMPLE

**Instruction :**

>    You need to issue an action,interaction history summary for this step.
>       When you thinking need have OBSERVATION DESCRIPTION, OBSERVATION
>       HIGHLIGHT, REASON.
>
>    You are ONLY allowed to use the following action commands. Strictly
>       adheres to the given format.
>
>    {click [id]: To click on an element with its numerical ID on the webpage.
>       E.g., `click [7]` ;
>
>    type [id] [content] [press_enter_after=0|1]: To type content into a field
>       with a specific ID. By default, the \"Enter\" key is pressed after
>       typing unless `press_enter_after` is set to 0. E.g., `type [15] [
>       Carnegie Mellon University] [1]` .;
>
>    stop [answer]: To stop interaction and return response. Present your
>       answer within the brackets. If the task doesn't require a textual
>       answer or appears insurmountable, additional reasons and all relevant
>        information you gather as the answer. E.g., `stop [N/A ...]`. ;
>
>    note [content]: To take note of all important info w.r.t. completing the
>       task to enable reviewing it later. E.g., `note [Spent $10 on
>       4/1/2024]`;
>
>    go_back: To return to the previously viewed page.

**Input :**

>    OBJECTIVE: Which US states border Vermont?
>
>    LASTACTION: first step, no last action
>
>    LAST INTERACTION HISTORY SUMMARY: This is the initial interaction. We are
>       on the OpenStreetMap homepage and need to search for Vermont to begin
>        our task of identifying its bordering states.
>
>    CURRENT OBSERVATION:
>        RootWebArea "OpenStreetMap"
>        link [41] 'OpenStreetMap logo OpenStreetMap'
>        link [402] 'Edit'
>        button [403]
>        link [373] 'History'
>        link [374] 'Export'
>        link [407] 'GPS Traces'
>        link [408] 'User Diaries'
>        link [409] 'Communities'
>        link [410] 'Copyright'
>        link [411] 'Help'
>        link [412] 'About'
>        link [382] 'Log In'
>        link [383] 'Sign Up'
>        link [515] 'Where is this?'
>        textbox [35] 'Search' [required: False]
>        button [516] 'Go'
>        link [503] 'Find directions between two points'
>        heading 'Welcome to OpenStreetMap!'
>        button [469] 'Close'
>        text 'OpenStreetMap is a map of the world, created by people like you
>            and free to use under an open license.'
>        text 'Hosting is supported by'

```
            link [475] 'UCL'
            link [477] 'Fastly'
            link [479] 'Bytemark Hosting'
            text ', and other'
            link [481] 'partners'
            link [485] 'Learn More'
            link [486] 'Start Mapping'
            generic [15] 'Zoom In Zoom Out Show My Location Layers Share 50 km 50
                 mi OpenStreetMap contributors Make a Donation. Website and API
                 terms'
            link [19] 'Zoom In'
            link [20] 'Zoom Out'
            button [22] 'Show My Location'
            link [24] 'Layers'
            link [296]
            link [26] 'Share'
            link [309] 'OpenStreetMap contributors'
            link [311] 'Make a Donation'
            link [313] 'Website and API terms'
```

**Output :**

```
    "<think>
    OBSERVATION DESCRIPTION: The current page is the OpenStreetMap
         homepage. It contains a search box (ID: 35) where we can enter our
         query. There are various navigation links and buttons, but the
         most relevant element for our task is the search functionality.
         The map view (ID: 15) is also present, which will be crucial for
         visualizing Vermont and its neighboring states once we perform the
         search.

    OBSERVATION HIGHLIGHT: 35, 15, 516, 19, 20, 24, 26

    REASON: To locate Vermont on the map, we need to use the search
         functionality.

    </think>
    ACTION: type [35] [Vermont] [1]

    INTERACTION HISTORY SUMMARY: We have searched for \"Vermont\" on
         OpenStreetMap and are now looking at the search results page with a
         map view of Vermont."
```

## J  DATASET STATISTICS

| Dataset | Domain | License | Train (Original) | Train (Filtered) | Test |
|---|---|---|---|---|---|
| SQA Geva et al. (2021) | Commonsense | MIT | 2,061 | 1,544 | 229 |
| CSQA Talmor et al. (2019) | Commonsense | MIT | 9,741 | 6,478 | 1,140 |
| ARC Clark et al. (2018) | Commonsense | CC BY-SA 4.0 | 1,199 | 1,035 | 1,172 |
| BoolQ Clark et al. (2019) | Commonsense | CC BY-SA 3.0 | 9,427 | 0 | 3,270 |
| OpenbookQA Mihaylov et al. (2018) | Commonsense | Apache | 4957 | 0 | 500 |
| MATH Hendrycks et al. (2021) | Math | MIT | 7,500 | 2,511 | 5,000 |
| GSM8K Cobbe et al. (2021) | Math | MIT | 7,379 | 4,293 | 1,339 |
| GSM8K-Rev Guo et al. (2024) | Math | Apache | - | 0 | 777 |
| ANLI (r3) Nie et al. (2020) | NLI | CC BY-NC 4.0 | 100,459 | 883 | 1,200 |
| e-SNLI Camburu et al. (2018) | NLI | CC BY-NC 4.0 | 549,367 | 0 | 9,824 |
| Date Srivastava et al. (2022) | Logic | Apache | - | 200 | 169 |
| Webarena Zhou et al. (2023) | Agentic | Apache | - | 0 | 812 |
| Visual-CoT Shao et al. (2024) | Visual | Apache | 132,000 | 943,000 | 12,500 |
| OK-VQA Marino et al. (2019) | Visual | CC BY 4.0 | 5,046 | 9,009 | 5,000 |

Table 11: The datasets used in our Experimental Setup.

## K  THE USE OF LARGE LANGUAGE MODELS (LLMS)

To enhance clarity and readability, we employed OpenAI's GPT-5 and GPT-5-thinking models exclusively as language polishing tools. Their role was limited to proofreading, grammatical correction, and stylistic refinement—functions comparable to those of conventional grammar checkers and dictionaries. These tools did not contribute any new scientific content or ideas, and their usage is consistent with standard practices in manuscript preparation.

