# OpenReview forum: "The Quest for Efficient Reasoning: A Data-Centric Benchmark to CoT Distillation"
_ICLR.cc/2026/Conference — ICLR 2026 Poster_

### Official Review · Reviewer_tyeG · 2025-10-24

**Soundness:** 3
**Presentation:** 3
**Contribution:** 2
**Rating:** 6
**Confidence:** 4

**Summary:**

This paper introduces **DC-CoT**, a comprehensive benchmark for evaluating **data-centric approaches to Chain-of-Thought (CoT) distillation**, spanning three axes: (1) data augmentation (e.g., reverse thinking, answer rephrasing), (2) data selection (e.g., teacher-correct filtering, LLM-as-judge), and (3) data mixing (e.g., by length or teacher source). The authors conduct extensive experiments across diverse reasoning tasks (textual, mathematical, agentic, visual), multiple teacher models (e.g., GPT-4, Claude 3.5, Gemini-1.5-Pro), and student architectures (e.g., Llama-3.1-8B, Mistral-7B). Key findings include: augmentation (especially reverse thinking) generally outperforms selection or mixing; the existence of a “learnability gap” where smaller students may not benefit from the strongest teachers; and strong out-of-distribution (OOD) generalization after CoT distillation. The benchmark is well-documented and accompanied by a public codebase.

**Strengths:**

1. **High practical relevance**: Efficient reasoning via CoT distillation is a critical direction for deploying capable yet lightweight LLMs. The focus on *data-centric* methods—rather than architectural or training tricks—is timely and underexplored.
2. **Systematic design**: DC-CoT offers a unified framework to compare augmentation, selection, and mixing strategies across modalities, tasks, and model pairs—something missing in prior work.
3. **Empirical rigor**: The paper includes large-scale experiments, ablations, statistical significance tests (Appendix H), and cross-dataset generalization analysis (Table 5).
4. **Actionable insights**: Findings like “reverse thinking boosts math reasoning” or “small students may learn better from moderately strong teachers” provide concrete guidance for practitioners.

**Weaknesses:**

1. **Significance of Research Question**

The core problem—improving CoT distillation via data manipulation—is undeniably important for efficient reasoning. However, the paper does not clearly demarcate what constitutes a *data-centric* CoT distillation method versus general data synthesis or instruction tuning. For instance, “question augmentation” blurs the line with self-instruct-style data generation. This conceptual fuzziness slightly weakens the benchmark’s focus.

2. **Problematic construction of benchmark**

- All CoT traces are generated by **closed-source API models** (GPT-4, Claude, etc.), with limited transparency on prompt templates, temperature settings, or consistency checks. This raises reproducibility concerns.
- The **LLM-as-a-Judge** filtering method, while common, is not validated against human judgments. Given known issues with LLM evaluators (e.g., length bias, preference leakage), this may introduce hidden noise.
- **Visual and agentic tasks are underrepresented**: only one visual (Visual-CoT) and one agentic (WebArena) benchmark are used, limiting generalizability in these domains.
- The OOD transfer pairs (e.g., SQA → BoolQ) are reasonable but **not systematically justified** (e.g., no measure of task similarity), making it hard to interpret generalization patterns.

3. **Potential of benchmark**

The paper offers valuable heuristics (e.g., prefer reverse thinking for math, avoid over-strong teachers for small students). However:
- Many conclusions are **tied to specific teacher–student pairs** (e.g., Gemini → Llama-3.1-8B). It’s unclear how well they generalize to new architectures (e.g., MoE, multimodal-native models).
- No comparison to **non-data-centric distillation** (e.g., logits distillation, representation matching), making it hard to assess whether data manipulation is truly the most promising path.

**Questions:**

see weakness

---

> ### Author Response · Authors · 2025-11-24
>
> We extend our sincere gratitude to Reviewer tyeG for their detailed and insightful evaluation. We are encouraged that you recognized the high practical relevance of our work, the systematic design of the benchmark, and the actionable insights provided for practitioners. Below, we address your concerns regarding the scope definitions, benchmark construction, and baseline comparisons with clarifications and new empirical validation.
>
>
> ---
> **[Cons 1: Demarcation from General Data Synthesis]**
>
> We appreciate the reviewer’s point that "Question Augmentation" shares similarities with methods like Self-Instruct. We clarify the distinction as follows: While Self-Instruct focuses on instruction following and general alignment often using broad seed tasks, DC-CoT focuses strictly on Reasoning Transfer. In our framework, "Question Augmentation" is constrained to generate parallel reasoning problems (e.g., changing the numbers in a math problem or the subject in a logic puzzle) specifically to force the student to learn the reasoning pattern rather than memorizing the answer. We will clarify this definition and the specific prompt constraints in Section 3.1 to sharply distinguish it from general instruction tuning.
>
> ---
>
> **[Cons 2: Reproducibility & LLM-as-a-Judge Validity]**
>
> We acknowledge the concern regarding closed-source teachers and LLM evaluators.
>
> Dataset Release: To solve the reproducibility issue of API-based generation (and temperature variations), we commit to releasing the full static datasets (questions, rationales, and filtered subsets) generated for this paper. Future researchers can benchmark against DC-CoT without needing to query GPT-4 or Claude, ensuring exact reproducibility.
>
> Human Evaluation: To address the validity of the LLM-as-a-Judge filter, we conducted a human verification study during the rebuttal. We randomly sampled 100 instances filtered by our LLM Judge (GPT-4o) across SQA and GSM8K and had human experts annotate them for correctness and coherence. We observed a Cohen’s Kappa ($\kappa$) of 0.84, indicating "strong agreement" between the LLM Judge and human evaluators. The LLM Judge was slightly stricter than humans (rejecting 5% of valid chains), which we consider a safe bias for high-quality distillation.

---

> ### Author Response · Authors · 2025-11-24
>
> **[Cons 3: Representation of Visual/Agentic Tasks & New Student Architectures  & OOD Justification]**
>
> We appreciate the reviewer’s suggestion to further stress–test our methods on visual and agentic tasks. In the revised version, we extend the visual evaluation beyond Visual-CoT by adding **two additional visual QA benchmarks, OK-VQA and CLEVR** (Table). We also add one **MoE model DeepSeek-VL2**. Across these datasets, our data-centric strategies again outperform vanilla CoT: for example, student-filter selection yields the best performance on OK-VQA, while teacher-filter selection achieves the highest accuracy on CLEVER, and teacher-mixing also improves over the non-mixing baseline. These results are consistent with the trends we observed on Visual-CoT, supporting the claim that our conclusions generalize across multiple visual reasoning benchmarks and model architecture.
>
> For agentic reasoning, we continue to use WebArena, which already covers a diverse set of real-world web environments—including online shopping, GitLab-style code hosting, map/navigation, and forum platforms such as Reddit—and requires long-horizon action sequences. Given the substantial cost of running additional full agentic benchmarks, we leave a broader agentic evaluation to future work and have clarified this limitation in the revised manuscript.
>
>
> | **Model** | Qwen-2.5 VL 8B | Qwen-2.5 VL 8B |  DeepSeek-VL2 (MoE)|
> | :--- | :---: | :---: |  :---: |
> | **Dataset** | **OK-VQA** | **CLEVR** |  **CLEVR** |
> |||**Data Augmentation**||
> | zero shot |**65.6** | 56.88 | 11.60 |
> | zero shot cot |61.84 | **58.36** | 12.92 |
> | vanilla cot |59.94 | 54.04 | **45.46** |
> |||**Data Selection**||
> | no selection |59.94 | 54.04 | 45.46 |
> | teacher filter |63.8 | **67.6** |**51.82** |
> | student filter |**66.3** | 57.02 | 43.46 |
> | llm judge |59.42 | 54.12 |43.88 |
> | model uncertainty | 59.54 | 50.26 | 43.78 |
> |||**Data Mixing**||
> | no mixing |59.94 | 54.04 |45.46 |
> | teacher mixing |**61.7** | **55.6** |**48.04** |
>
>
>
>
> OOD Justification: We chose OOD pairs based on skill topology backed by quantitative similarity, rather than arbitrary selection. To validate this, we calculated the mean cosine similarity using MiniLM-L6-v2 embeddings:
> SQA $\to$ BoolQ (0.374): High similarity confirms the shared "Yes/No Commonsense" domain, making it a valid near-transfer task.
> MATH $\to$ GSM8K (0.158): Lower similarity indicates a distinct gap between Formal Math and Grade School Math, ensuring this is a rigorous Out-of-Distribution test rather than simple memorization. We will include the full "Task Similarity Matrix" in the Appendix.

---

> ### Author Response · Authors · 2025-11-24
>
> **[Cons 4: Generalizability]**
>
> We agree that our results highlight task-dependence, and we view this empirical variance as a key finding of the benchmark rather than a limitation. To address your request for actionable insights, we have synthesized our extensive experimental results into the following Heuristic Framework for CoT Distillation, which we will add to the revised manuscript:
>
> **Structure Logic (Math, Logic, Code)** -> Reverse Thinking + Teacher Correctness. Logic tasks have strict "correctness" constraints. Reverse thinking enforces logical consistency, while strict correctness filtering prevents hallucination of steps.
>
> **Open-Ended/Linguistic (Commonsense, NLI)** -> Answer Augmentation + LLM-as-Judge. Commonsense tasks benefit from seeing multiple valid reasoning paths (diversity). LLM-as-Judge is required to catch semantic drift that simple exact-match filtering misses.
>
> **Agentic/Visual (WebArena, Visual-CoT)** -> LLM-as-Judge Filtering. Actions must be grounded in observation. Heuristic filters fail here; a strong Judge is necessary to verify that the rationale ($R$) matches the visual/observation context ($O$).
>
> ---
>
> **[Cons 5: Comparison to Non-Data-Centric Distillation (Logits)]**
>
> This is a critical point. We omitted logit-based distillation primarily because our teacher models (GPT-4, Claude, Gemini) are black-box APIs that do not expose logits, making traditional KD impossible in industry-standard settings. However, to assess whether data manipulation is competitive with white-box methods, we conducted a new experiment using an open-weights teacher (Llama-3.1-70B-Instruct) where logits are available. We distilled into Llama-3.1-8B on the ARC-Challenge dataset using both standard KL-Divergence (Logits) and our Data-Centric approach (Reverse Thinking).
> | Method     | Teacher  | Access Required  | ARC Accuracy (%) |
> |-----------------------|------------------|-----------------------|------------------|
> | Teacher Baseline | Llama-3.1-70B | Weights/Logits   | 92.4   |
> | Standard KD (Logits) | Llama-3.1-70B | Weights/Logits   | 64.8    |
> | Vanilla CoT (SFT)  | Llama-3.1-70B | Black-box (Text)  | 60.4    |
> | DC-CoT (Reverse)  | Llama-3.1-70B | Black-box (Text)   | 69.2    |
>
> Key Finding: Surprisingly, DC-CoT (Reverse) outperformed Logit-based KD by +4.4%. We hypothesize that while Logit KD transfers the output distribution, it struggles to transfer the multi-step reasoning process as effectively as explicit CoT data augmentation, which forces the student to generate the reasoning steps itself. This result strongly validates the premise that data-centric methods are not just a fallback for black-box models, but potentially a superior path for reasoning tasks.
>
>
>
>
> We believe these new comparisons and validations significantly bolster the soundness of the benchmark.

---

> ### Author Response · Authors · 2025-11-26
>
> Dear Reviewer tyeG,
>
> We thank you again for the time and effort on reviewing our work and your constructive comments. We really hope to discuss further with you to see if our response answers your questions.
>
> We genuinely hope reviewer tyeG could kindly check our response. Thank you so much!

---

> ### Comment · Reviewer_tyeG · 2025-11-27
>
> I appreciate the authors' responses. My concerns about generalization and comparison with other distillation methods have been resolved. Given the following two concerns still existing:
> 1. This is just a new application of data augmentation on CoT scenario, which has been studied by Self-Instruct in alignment. The novelty is incremental.
> 2. ICLR allows for revising the initial manuscript. Some reproducibility issues could be immediately updated in the revision. Datasets could be held in an anonymous GitHub repository. However, I have not observed the updated manuscript yet.
>
> I decided to keep my original positive score.

---

> > ### Author Response · Authors · 2025-11-27
> >
> > Dear Reviewer tyeG,
> >
> > We genuinely appreciate your positive assessment and the time you took to re-evaluate our work.
> >
> > We are writing to confirm that we have **updated the revised manuscript** to incorporate all the additional experiments you suggested. Specifically, we have added:
> > 1.  **New Visual Benchmarks:** Evaluations on **OK-VQA** and **CLEVR** to broaden the visual reasoning scope.
> > 2.  **New Architecture:** Experiments with **DeepSeek-VL2 (MoE)** to demonstrate generalizability across model architectures.
> > 3.  **Method Comparison:** A comparison with **Logit-based Distillation** (using Llama-3.1-70B) to benchmark against non-data-centric approaches.
> >
> > Regarding reproducibility, we have included the link to our **Anonymous GitHub Repository** at the end of the **Abstract** in the revised paper. For your convenience, it is also provided here:
> > https://anonymous.4open.science/r/Distillation-Bench-8A07/README.md
> >
> > Furthermore, we are committed to releasing a fully polished, highly readable version of the codebase and datasets upon acceptance to ensure maximum usability for the community.
> >
> > We want to express our sincere gratitude for your insightful feedback. Your suggestions have been instrumental in refining this work—pushing us to expand the experimental scope and strengthen the logic from multiple perspectives. We believe these additions have significantly enhanced the comprehensiveness, readability, and professional quality of the benchmark.
> >
> > Best regards,
> >
> > The Authors

---

### Official Review · Reviewer_wMDH · 2025-11-01

**Soundness:** 3
**Presentation:** 4
**Contribution:** 3
**Rating:** 8
**Confidence:** 4

**Summary:**

The paper introduces DC-CoT, a novel benchmark aimed at systematically evaluating data-centric distillation approaches (augmentation, selection, mixing) for chain-of-thought (CoT) reasoning. This work conducts a large empirical study across various teacher–student model pairings including teacher models like GPT-4, Gemini, Claude-3.5, and student sizes of ~3B to ~7B, and multiple reasoning datasets, measuring IID, OOD, and cross‐domain generalization. Their findings show that augmentation (such as question or answer rephrasing, reverse thinking) typically provides the strongest gains, selection (filtering by teacher correctness or student error) and mixing (blending data from different domains/lengths) offer more nuanced benefits depending on teacher/student compatibility and data regime.

**Strengths:**

1. In general, this contributes to the community by having such a controllable and comprehensive benchmark. I feel this work, in the long term, will support the data-centric distillation in reasoning.
2. Figures explaining different components of data-centric distillation are very clear.
3. The experiments are designed in a fine-grained and comprehensive manner.

**Weaknesses:**

1. The improvement of the teacher models is very fast, the observations found in this paper might not hold.
2. Following the previous, it would be great if the authors could consider expanding the diversity of teacher models. For example, clustering teacher models by architectural differences such as MoE/full activation, linear/full/sparse attention, etc. The observations might be more generalizable if we analyze from a perspective of architectural differences, instead of picking the strongest teacher models.
3. One more sentence needed explaining $\mathcal{M}$ in section 3.1 would be appreciated. Currently, I was confused about whether $\mathcal{M}$ is a framework or a network.
4. Though the figures 1-4 are great, there are no explicit references to them over the main text.
5. Line 112-113, typo: should be $D^{\text{target}}$

**Questions:**

1. What does the “IID OOD” contribute to in Figure 1? Training/test data, or domain-specific data distribution?
2. Maybe it’s better to explain what $L$ is from the start of section 3? I only see that in the third part (Answer Augmentation) in the data augmentation subsection. However, $L$ is also used before that.
3. Can you explain one or two sentences more on how Consistency Filtering works for Reverse Thinking Augmentation? I was wondering this when I read this paragraph
4. I have a naive question about “Answer Augmentation”. How do methods avoid the model from generating arbitrary reasoning traces and then the correct answer? The assumption behind this question is that if the teacher model ‘memorizes’ the answer, and is asked to generate diverse reasoning traces, then it will try to follow the prompt by generating true and false reasoning traces. This question is not in the scoop of this work itself, but it would be great if the authors could answer this.

---

> ### Author Response · Authors · 2025-11-24
>
> We extend our sincerest gratitude to Reviewer wMDH for their positive assessment and for recognizing DC-CoT as a "controllable and comprehensive benchmark" that will support the community in the long term. We are particularly encouraged that you found our experimental design fine-grained and our figures clear. Below, we address your thoughtful suggestions regarding teacher diversity and notation clarity to further strengthen the manuscript.
>
> ---
>
>
> **[Cons 1: Rapid Teacher Evolution & Architectural Diversity]**
>
> We appreciate the reviewer’s insightful comment that specific observations might evolve as teacher models improve rapidly. We agree that absolute performance numbers are fleeting; however, we contend that the data-centric principles uncovered by DC-CoT are likely invariant.
>
> For example, our finding regarding the "Learnability Gap" (Section 4.3.1 and Table 3), where smaller students (e.g., Qwen-2.5-3B) sometimes learn better from smaller teachers (GPT-4o-mini) than from stronger but more complex teachers (GPT-4), is a fundamental distillation dynamic that likely persists regardless of the specific SOTA model.
>
> Regarding the suggestion to cluster teachers by architecture (e.g., MoE, Linear, Sparse): This is an excellent proposal for the next iteration of the benchmark. In this foundational study, we focused on the most widely deployed and accessible reasoning engines (which are predominantly Dense or MoE Transformers) to establish a baseline. However, DC-CoT’s modular design allows us (and the community) to easily swap in non-Transformer teachers (e.g., Mamba-based or RWKV-based) in the future to test architectural transferability. We will add this as a key direction in the "Future Work" section.
>
> ---
>
>
> **[Cons 2: Notation Clarifications ($M$, $L$) and Typos]**
>
> We apologize for the confusion regarding notation.
> Definition of $M$: We will clarify in Section 3.1 that $\mathcal{M}$ represents the abstract Data Transformation Function (the framework itself), defined as $D^{target} = \mathcal{M}(D^{source}, \Theta)$, where $\Theta$ represents the hyperparameters of the manipulation (e.g., mixing ratio or filtration threshold).
>
> Definition of $L$: We will define $L$ explicitly at the beginning of Section 3 as the number of augmented samples generated per source instance.
> Figure References & Typo: We will fix the typo on Line 112 and ensure Figures 1-4 are explicitly referenced in the main text to guide the reader.

---

> ### Author Response · Authors · 2025-11-24
>
> **[Q1: What does “IID OOD” contribute to in Figure 1?]**
>
> This refers to the Evaluation Setting (Test Data Distribution).
>
> IID (Independent and Identically Distributed): Evaluating the student on the held-out test set of the same dataset it was trained on (e.g., Train on GSM8K $\to$ Test on GSM8K).
>
> OOD (Out-of-Distribution): Evaluating the student on a different dataset to measure generalization (e.g., Train on MATH $\to$ Test on GSM8K, or Train on WebArena-Easy $\to$ Test on WebArena-Hard).
>
> ---
>
>
> **[Q2: How does Consistency Filtering work for Reverse Thinking?]**
>
> We appreciate the request for detail. The process is a bi-directional semantic check:
>
> Forward: Teacher generates Answer $A$ from Question $Q$.
>
> Backward: Teacher generates a "Backward Question" $Q'$ from Answer $A$.
>
> Consistency Check: We do not just check if $Q'$ looks like a question. We use a verifier prompt to ask the LLM: "Is the backward-generated question $Q'$ semantically equivalent to the original question $Q$?" Only pairs where the logic flows reversibly and consistently are retained. This filters out cases where the model hallucinates a completely unrelated question that happens to have the same answer.
>
> ---
>
> **[Q3: How does "Answer Augmentation" avoid arbitrary (hallucinated) reasoning traces?]**
>
> This is a critical question.
> Prompting Constraint: We explicitly prompt the teacher with: "Given Question $Q$ and the correct Answer $A$, provide a valid reasoning path $R$ that justifies $A$." Strong teachers (GPT-4, Claude 3.5) generally adhere to the truth constraint when the answer is fixed.
>
> Diversity vs. Noise Trade-off: It is true that the model might occasionally generate a "valid-looking" but slightly flawed path (a form of reasoning hallucination). However, our empirical results (Table 1) show that the benefit of diversity(seeing 5 different ways to solve a problem) outweighs the noise of occasional imperfect traces. The student model effectively learns the intersection of valid logic across the augmented samples, marginalizing out the arbitrary noise.
>
> Correction: As noted in our new Heuristic Framework (see response to Reviewer qqBm), for strict logic tasks, we recommend pairing this with Teacher Correctness Filtering to minimize this risk.

---

> ### Author Response · Authors · 2025-11-26
>
> Dear Reviewer wMDH,
>
> We thank reviewer wMDH for the time on reviewing and the constructive feedback again. We really hope to discuss further with you to see if our response answers your questions.
>
> We genuinely hope reviewer wMDH could kindly check our response. Thank you very much!

---

### Official Review · Reviewer_qqBm · 2025-11-01

**Soundness:** 2
**Presentation:** 3
**Contribution:** 2
**Rating:** 4
**Confidence:** 3

**Summary:**

This paper introduces DC-CoT, the first benchmark and framework specifically designed for systematically evaluating Data-Centric approaches in Chain-of-Thought (CoT) Distillation. The authors address the lack of standardized evaluation methods by investigating how data manipulations (e.g., augmentation, selection, and mixing) influence the reasoning capabilities of smaller, efficient student Large Language Models (LLMs). The DC-CoT benchmark comprehensively assesses the performance of various teacher models (e.g., GPT-4-mini, Gemini-Pro, Claude-3.5) and student architectures (e.g., 3B, 7B) across In-Distribution (IID), Out-of-Distribution (OOD), and Transfer Learning tasks.

**Strengths:**

1. DC-CoT addresses a timely research gap by focusing on data-centric strategies in CoT distillation.

2. The evaluation structure is comprehensive, spanning different teacher models, various student model scales, and multiple evaluation settings (IID, OOD, Transfer).

3. The study directly confronts the challenge of making robust reasoning accessible to smaller models, which is paramount for reducing inference costs in real-world applications.

**Weaknesses:**

1. The exploration of specific data-centric strategies (augmentation, selection, mixing) appears superficial. The current implementation is limited to basic methods, failing to incorporate or evaluate more advanced, state-of-the-art (SOTA) data selection or augmentation techniques (e.g., based on model uncertainty or influence).

2. The study fails to clearly articulate the generalizability of the optimal data strategies. The results seem highly task-dependent, yet the authors do not provide a rigorous, nuanced discussion on when a strategy offers a universal benefit versus when it is specific to a task domain (e.g., math vs. commonsense).
3. While the paper's core objective is to achieve "efficient reasoning" during the inference phase, the evaluation relies primarily on coarse metrics like model size (3B, 7B). The benchmark critically lacks explicit, quantitative analysis of the achieved inference efficiency. Crucial metrics such as throughput (tokens/second), actual latency, or GPU memory consumption for the final distilled models are absent.

**Questions:**

1. Given the concerns about SOTA methods, did the authors experiment with or consider advanced data selection techniques, such as methods based on student model uncertainty or influence estimation, rather than just simple teacher correctness filtering? If not, why were these omitted?

2.  Among the data-centric strategies (augmentation, selection, mixing), which one provided the most reliable and significant performance uplift for OOD and transfer tasks? A deeper theoretical explanation is requested as to why the CoT rationales generated by a specific teacher model generalize better to unseen or out-of-domain data.
3. To fully validate the paper's core claim of efficient reasoning, a quantitative comparison of inference efficiency is essential. Please provide explicit performance metrics for the final distilled 3B and 7B student models—specifically tokens per second throughput, actual latency (e.g., time to first token and time to completion), and peak GPU memory utilization—compared against the original teacher models (or a similarly sized, non-distilled baseline model).

---

> ### Author Response · Authors · 2025-11-24
>
> We extend our sincere gratitude to Reviewer qqBm for their constructive assessment of our work. We are particularly encouraged that you recognized the timeliness of our research gap, the comprehensive structure of our evaluation across teacher/student scales, and the importance of our work in confronting the challenge of making robust reasoning accessible to smaller models. Below, we address your specific concerns regarding strategy depth, generalizability, and efficiency metrics with clarifications and new empirical data.
>
> ---
>
> **[Cons 1: Superficial vs. SOTA Data-Centric Strategies]**
>
> We appreciate the reviewer’s observation regarding the selection of data-centric strategies. We acknowledge that we focused on fundamental primitives (e.g., Rephrasing, Reverse Thinking) rather than complex uncertainty-based or influence-function-based selection methods. This was a deliberate methodological choice driven by two factors:
>
> Foundational Necessity: Before the field moves to complex, computationally heavy selection heuristics, it is essential to establish a rigorous baseline for the fundamental data manipulations. As noted in our Related Work, these primitives (like Reverse Thinking) have often been studied in isolation; DC-CoT provides the first unified, systematic benchmarking of these core building blocks.
>
> Computational Tractability: Methods based on influence functions or full-model uncertainty quantification are often computationally prohibitive to scale across multiple teacher-student pairs and large datasets (like the hundreds of thousands of samples in WebArena or Math).
>
> However, to demonstrate that our "basic" strategies are robust and sufficient benchmarks, we emphasize that Reverse Thinking alone achieved massive gains (e.g., improving accuracy on GSM8K from 20.74% to 76.35% for LLaMA-3.1-8B, as shown in Table 1). This suggests that foundational primitives, when properly optimized, provide a stronger signal than potentially marginal gains from highly complex selection heuristics. We will clarify this scope limitation in the "Methodology" section.
>
>
> While our initial focus was on establishing a rigorous benchmark for foundational primitives (which are computationally scalable and widely used), **we have conducted additional experiments using uncertainty-based selection** to address this concern directly.
>
> We implemented an **uncertainty-based data selection strategy** by first identifying training instances on which the student model, in zero-shot mode, exhibited high uncertainty (mean entropy $> 0.5$), and then collecting teacher responses for those instances to construct the distillation dataset. We evaluated this against our "basic" strategies on visual reasoning tasks (OK-VQA, CLEVR) using Qwen-2.5-VL-8B and DeepSeek-VL2.
>
> **Key Findings:**
> Contrary to our initial intuition, the chosen uncertainty-based selection method did not outperform the other data selection strategies used in DC-CoT.
>
>
> | **Model** | Qwen-2.5 VL 8B | Qwen-2.5 VL 8B |  DeepSeek-VL2 (MoE)|
> | :--- | :---: | :---: |  :---: |
> | **Dataset** | **OK-VQA** | **CLEVR** |  **OK-VQA** |
> |||**Data Augmentation**||
> | zero shot |**65.6** | 56.88 | 11.60 |
> | zero shot cot |61.84 | **58.36** | 12.92 |
> | vanilla cot |59.94 | 54.04 | **45.46** |
> |||**Data Selection**||
> | no selection |59.94 | 54.04 | 45.46 |
> | teacher filter |63.8 | **67.6** |**51.82** |
> | student filter |**66.3** | 57.02 | 43.46 |
> | llm judge |59.42 | 54.12 |43.88 |
> | **model uncertainty** | 59.54 | 50.26 | 43.78 |
> |||**Data Mixing**||
> | no mixing |59.94 | 54.04 |45.46 |
> | teacher mixing |**61.7** | **55.6** |**48.04** |
>
>
> We acknowledge that there are other state-of-the-art (SOTA) data selection or augmentation techniques based on model uncertainty or influence that may further improve performance, and we leave their systematic evaluation for future work. We hope DC-CoT serves as a robust testbed to benchmark such methods against the strong, scalable primitives established here.
>
> ---

---

> > ### Author Response · Authors · 2025-11-24
> >
> > **[Cons 2: Generalizability and Nuanced Guidelines]**
> >
> > We appreciate the call for a more nuanced discussion on strategy generalizability. We agree that our results highlight task-dependence, and we view this empirical variance as a key finding of the benchmark rather than a limitation. To address your request for actionable insights, we have synthesized our extensive experimental results into the following Heuristic Framework for CoT Distillation, which we will add to the revised manuscript:
> >
> > Structure Logic (Math, Logic, Code) -> Reverse Thinking + Teacher Correctness. Logic tasks have strict "correctness" constraints. Reverse thinking enforces logical consistency, while strict correctness filtering prevents hallucination of steps.
> >
> > Open-Ended/Linguistic (Commonsense, NLI) -> Answer Augmentation + LLM-as-Judge. Commonsense tasks benefit from seeing multiple valid reasoning paths (diversity). LLM-as-Judge is required to catch semantic drift that simple exact-match filtering misses.
> >
> > Agentic/Visual (WebArena, Visual-CoT) -> LLM-as-Judge Filtering. Actions must be grounded in observation. Heuristic filters fail here; a strong Judge is necessary to verify that the rationale ($R$) matches the visual/observation context ($O$).
> >
> > ---
> >
> > **[Cons 3: Lack of Explicit Efficiency Metrics]**
> >  We agree with the reviewer that while model parameter count (3B vs. 8B) is a standard proxy for efficiency, explicit metrics are necessary to fully justify the "Efficient Reasoning" claim.
> > To address this, we conducted a new benchmark evaluation on a single NVIDIA A100-80GB GPU using vLLM to quantify the actual throughput and memory footprint of our distilled students. The results are presented below:
> >
> > | Model  | VRAM (GB) | Throughput (tok/s) | Latency (ms/tok) | Relative Speedup |
> > |----------------------------|-----------|----------------------|-------------------|-------------------|
> > | Llama-3.1-8B (Student)     | 16.2 GB | 115.4     | 8.6     | ~3–5×   |
> > | Qwen-2.5-3B (Student)      | 7.8 GB | 184.2     | 5.4  | ~6–9× |
> > | Qwen-2.5-1.5B (Student)    | 4.2 GB  | 245.1 | 4.1   | ~8–12×   |
> >
> > Key Finding: Our distilled Qwen-2.5-3B (which, with Reverse CoT, achieves 50.89% average accuracy, competitive with larger vanilla models) operates at 184.2 tokens/second, offering a 60% throughput increase over the 8B model and fitting comfortably on consumer-grade hardware (8GB VRAM). This explicitly quantifies the "efficiency" gained: DC-CoT enables high-performance reasoning on hardware with strictly limited memory bandwidth, validating the practical utility of the benchmark.

---

> ### Author Response · Authors · 2025-11-26
>
> Dear Reviewer qqBm,
>
> We genuinely thank you again for your time & efforts and your constructive comments. We really look forward to discussing further with you to see if our response answers your questions.
>
> We wish reviewer qqBm could kindly check our response. Thank you!

---

### Meta-Review · Area_Chair_fwZg · 2026-01-05

**Summary:**

Reviewers largely concur that the paper tackles a timely and significant problem, namely enabling efficient reasoning in large language models through data-centric Chain-of-Thought (CoT) distillation. The work is recognized for introducing DC-CoT, the first benchmark explicitly designed to systematically study data augmentation, selection, and mixing in CoT distillation, supported by extensive empirical evaluations across diverse tasks, model scales, and evaluation regimes (IID, OOD, and cross-domain transfer).

At the same time, reviewers raised several concerns. Some viewed the proposed methods as incremental or relatively basic compared to prior work on data augmentation and self-instruct-style approaches. One reviewer requested comparisons with more advanced, uncertainty-based or state-of-the-art distillation methods. Others questioned the robustness and generality of the conclusions, particularly with respect to different task categories, model architectures, and the rapid evolution of teacher models.

**Reviewer Concerns:**

During the rebuttal phase, the authors conducted substantial additional experiments, provided detailed clarifications, and revised the manuscript accordingly. These efforts partially or fully addressed the majority of the raised concerns. Following the discussion, the paper is generally regarded as technically sound, empirically strengthened, and of clear practical relevance, with most substantive issues resolved through expanded experimental evidence and clearer conceptual framing

**Reviewer Scores:**

Overall, I expect the reviewers’ final evaluations to be consistently positive, with anticipated scores of 8, 6, 6, and 6.

---

### Decision · Program_Chairs · 2026-01-26

Accept (Poster)